# Failed Neuroprotection of Combined Inhibition of L-Type and ASIC1a Calcium Channels with Nimodipine and Amiloride

**DOI:** 10.3390/ijms21238921

**Published:** 2020-11-24

**Authors:** Jonas Ort, Benedikt Kremer, Linda Grüßer, Romy Blaumeiser-Debarry, Hans Clusmann, Mark Coburn, Anke Höllig, Ute Lindauer

**Affiliations:** 1Department of Neurosurgery, Medical Faculty, RWTH Aachen University, 52074 Aachen, Germany; bkremer@ukaachen.de (B.K.); hclusmann@ukaachen.de (H.C.); ahoellig@ukaachen.de (A.H.); ulindauer@ukaachen.de (U.L.); 2Department of Anaesthesiology, Medical Faculty, RWTH Aachen University, 52074 Aachen, Germany; lgruesser@ukaachen.de (L.G.); rblaumeiser-debarry@ukaachen.de (R.B.-D.); 3Department of Anaesthesiology and Intensive Care Medicine, University Hospital Bonn, 53127 Bonn, Germany; mark.coburn@ukbonn.de

**Keywords:** neuroprotection, neural injury, nimodipine, subarachnoid haemorrhage, acid-sensing ion channels, amiloride, oxygen-glucose deprivation

## Abstract

Effective pharmacological neuroprotection is one of the most desired aims in modern medicine. We postulated that a combination of two clinically used drugs—nimodipine (L-Type voltage-gated calcium channel blocker) and amiloride (acid-sensing ion channel inhibitor)—might act synergistically in an experimental model of ischaemia, targeting the intracellular rise in calcium as a pathway in neuronal cell death. We used organotypic hippocampal slices of mice pups and a well-established regimen of oxygen-glucose deprivation (OGD) to assess a possible neuroprotective effect. Neither nimodipine (at 10 or 20 µM) alone or in combination with amiloride (at 100 µM) showed any amelioration. Dissolved at 2.0 Vol.% dimethyl-sulfoxide (DMSO), the combination of both components even increased cell damage (*p* = 0.0001), an effect not observed with amiloride alone. We conclude that neither amiloride nor nimodipine do offer neuroprotection in an in vitro ischaemia model. On a technical note, the use of DMSO should be carefully evaluated in neuroprotective experiments, since it possibly alters cell damage.

## 1. Introduction

Despite years of research pharmacological neuroprotection remains challenging. Neuronal cell death represents the smallest common denominator occurring within the pathophysiological cascade of acute cerebrovascular or traumatic brain diseases. For example, subarachnoid haemorrhage occurs with an incidence of 9/100,000 per year [1], while traumatic brain injury is one of the leading causes for death and disability in the young with reported incidences of 262/100,000 per year [2]. Considering the dreads of brain damage for patients and their dependants, therapeutic means of effective neuroprotection is possibly one of the most aimed-for means in modern medicine.

Although the pathomechanisms of neuronal cell death are yet to be fully understood, one of the first and most frequently investigated events is the elevation of intracellular calcium concentration. Elevated intracellular calcium represents a central part within the early phenomenon of excitotoxicity as well as all the pathways resulting in or from these raised ion concentrations [3,4,5,6,7]. Various drugs blocking the excitotoxicity-induced calcium influx have failed to show effective neuroprotection in the first place or failed in overcoming the translational roadblock from in vivo or animal in vitro experiments to actual clinical application (e.g., trials of glutamate-receptor antagonists) [8,9]. Beside calcium influx via glutamate receptor overactivation, calcium influx may also occur via voltage-gated calcium channels (VGCC) located within neuronal membranes. Additionally, VGCCs are also expressed by vascular smooth muscle cells responsible for intracellular calcium rise and vasoconstriction when activated.

The L-type calcium channel inhibitor nimodipine is well established as an oral agent for the management of delayed cerebral ischaemia (DCI) for patients after subarachnoid haemorrhage (SAH) [10,11,12,13,14,15]. Although nimodipine induces vascular smooth muscle relaxation [16] and hence is widely believed to reduce vasospasms, research has shown that the beneficial effects in SAH patients with DCI may not be primarily caused by this effect on larger cerebral vessels [12,17], but possibly by a direct neuroprotective effect. Experimental studies have been rather controversial with some arguing towards direct neuroprotection [18,19,20,21] and some against [22,23,24,25]. In brief, the inhibition of voltage-gated L-type calcium channels is believed to provide protection against raised intracellular calcium concentration once the cell depolarises in ischaemic conditions. However, it remains unclear whether this effect also counts for the observed beneficial effect of VGCC inhibitors in SAH.

Other possible drug targets regarding cerebral ischaemia are sodium- and (to a lesser extent) calcium-permeable acid-sensing ion channels (ASICs), with the ASIC1a subtype channel as the most prominent amongst them [26]. ASIC1a channels can be responsible for neuronal damage in acidosis and provide a neither voltage-gated nor glutamate-dependent mechanism for calcium influx. Amiloride can block this non-specifically [27,28,29]. Many experiments have already shown that neuroprotection can be achieved by blocking ASIC1a and thus targeting these channels might be promising [9,30,31]. Some studies point out connections between the affinity of ASIC for H^+^ depending and the extracellular concentration of calcium [32,33], as it occurs in brain ischemia [34].

In our experiment, we investigated the possible neuroprotective effect of two clinically used drugs, which both block calcium channels: The L-type calcium channel inhibitor nimodipine and the acid-sensing ion channel (ASIC1a) blocker amiloride.

We hypothesised a possible synergistic neuroprotective effect of the two compounds being used in combination since ASIC channels are known to be influenced by external and internal calcium levels [32,33]. To assess our postulation, we used a well-established in vitro model of oxygen–glucose deprivation (OGD) in organotypic hippocampus slices of mice with propidium iodide (PI) staining for cell death assessment. Nimodipine was investigated for neuroprotective effects alone and in combination with amiloride to observe a possible synergistic interaction.

## 2. Results

### 2.1. OGD Damage

Comparing our control group (*n* = 102) with our OGD group (*n* = 96) a significant difference was seen (*p* < 0.0001, Mann–Whitney test), thus demonstrating significant and robust cell damage by our OGD regimen (Figure 1).

### 2.2. Effect of DMSO as Vehicle on OGD-Induced Damage

To rule out possible effects of our vehicle, slices were incubated receiving dimethyl-sulfoxide (DMSO) without any drug in addition either as controls (Figure 2, left) or undergoing OGD (Figure 2, right) at levels of 0.1, 1.0, or 2.0 Vol.% DMSO. For pure vehicle control slices without OGD, there was a tendency towards a concentration dependent effect; however, with significant cell impairment observed only with the highest concentration of 2.0 Vol.% DMSO (*p* = 0.0001, Kruskal–Wallis with post hoc Dunn’s test). Following OGD, there was a slight albeit far not significant tendendy towards a protective effect of DMSO (Figure 2, right: DMSO concentration of 1.0 Vol.%, *p* = 0.2237) compared with the OGD slices without vehicle or with other DMSO concentrations.

### 2.3. Effect of Nimodipine at 10 or 20 µM Dissolved in Varying DMSO Concentrations

L-type calcium channel blockage with nimodipine at either 10 or 20 µM dissolved in 0.1 or 1.0 Vol.% DMSO, respectively, did not show significant neuroprotection (Figure 3).

### 2.4. Effect of Nimodipine at 10 or 20 µM in Combination with 100 µM Amiloride Dissolved in Varying DMSO Concentrations

To observe a neuroprotective effect of combined blockage of L-type calcium channels and ASIC1a, a combination of 10 or 20 µM nimodipine and 100 µM amiloride dissolved in 1.1 Vol.% DMSO was applied. No significant effect for the combination of nimodipine and amiloride or amiloride alone at 1.0 Vol.% DMSO could be observed (Figure 4). Further, no significant difference was detected in slices receiving nimodipine at 10 or 20 µM in 1.0 Vol.% DMSO or with the vehicle alone. These slices showed a tendency towards less severe cell damage when compared to the OGD group and slices receiving regimens with 0.1 Vol.% DMSO (also Figure 4).

Lastly, treatment combination of 100 µM amiloride and either 10 or 20 µM nimodipine was investigated while using a final concentration of 2.0 Vol.% DMSO. No combination of nimodipine and amiloride or amiloride alone showed any significant neuroprotective effect. However, slices receiving the combination of 100 µM amiloride and 10µM (*p* = 0.0001, Kruskal–Wallis with post hoc Dunn’s test, Cohen’s d = 1.196243) or 20 µM (*p* = 0.0001, Kruskal–Wallis with post hoc Dunn’s test, Cohen’s d = 1.219638) nimodipine at 2.0 Vol.% DMSO displayed significantly higher levels of cell damage at 72 h after OGD compared to the OGD group without any treatment. (Figure 5).

## 3. Discussion

In our study, we investigated the clinically commonly used drugs nimodipine and amiloride to test for neuroprotective effects in an in vitro model of brain ischemia using OGD. No protective effect could be observed for nimodipine or amiloride alone or in combination. Our model was already used in the past to prove neuroprotective effects of other drugs, such as the noble gas argon, which showed a protection of >80% in the best suited protocol [35]. DMSO, which was used as vehicle to dissolve both drugs, induced significant impairment of cell integrity at 2.0 Vol.% in our control slices (that did not undergo OGD) compared with the control group without DMSO. Paradoxically, this effect was not observed at the same concentration in slices undergoing OGD. There was even a tendency towards ameliorated levels of cell damage in OGD slices at 1.0 Vol.% DMSO (Figure 2, right). Interestingly, the combination of nimodipine and amiloride significantly increased cell damage when dissolved at 2.0 Vol.% DMSO. This effect was not apparent for 2.0 Vol.% DMSO alone or with the same concentration as vehicle for 100µM amiloride. Nimodipine alone in 2.0 Vol.% DMSO was not investigated.

### 3.1. The Role of Calcium for Cell Death and Failure to Provide Neuroprotection

The exact mechanisms of neuronal cell death in ischemia are yet to be fully understood. Dirnagl et al. postulated in 1999 that there are several main mechanisms: excitotoxicity (i.e., damage through an overshooting release of neurotransmitters, mainly glutamate, followed by intracellular calcium overload), oxidative stress, cortical spreading depolarisations, inflammation, and apoptosis [36]. Calcium influx has long been labelled as “final common pathway” [3] of toxic cell death; however, this term might be misleading, since calcium itself activates a cascade of intracellular reactions and thus the possible mechanisms are manifold [37], e.g., a two-step model was suggested consisting of neuronal swelling in a first step and a delayed entrance of calcium in a second [23].

The main hypothesis of our study was that neuroprotection can be achieved by preventing the intracellular rise in calcium concentration by a combined blockade of two postulated calcium entry pathways, while blocking the L-Type calcium channel with nimodipine and the ASIC1a channel using amiloride.

Nimodipine has been reported to improve functional outcome in ischaemic and haemorrhagic stroke in in vivo animal experiments as well as in clinical trials [13,18,20,38,39] and is still the only therapeutic option in the treatment of DCI after SAH [40]. However, the effect of L-type channel inhibition might primarily be targeted to smooth muscle cell function to prevent large as well as small vessel contractions. A possible additional effect on neurons (and astrocytes) might be overrated. To separate direct neuroprotective from vascular effects, models without the dependency of blood circulation (such as cell culture or slice culture models) have been used to investigate this postulated effect of nimodipine and other VGCC inhibitors. Although there are data showing that nimodipine decreases membrane depolarization and thus calcium influx in OGD [21], Kass et al. for instance pointed out that it is rather the loss of calcium extrusion mechanisms (i.e., the dysfunction of the Ca^2+^-ATPase and the Na^+^-Ca^2+^-Exchanger due to the loss of ATP in hypoxic condition [41]) and not an increased calcium influx that is responsible for cell damage [42]. In addition, the question remains how relevant the portion of the L-type channel for calcium overload within ischaemic damage is. Some data suggest Q-type and N-type channels are far more critical with regard to neuronal injury and that the L-type channel accounts for less than 10% of total damage [24]. Most of the intracellular calcium overload appears to be the result of excitotoxic activation of N-methyl-D-aspartate (NMDA) receptors that become permeable for the ion [41,43]. Moreover, only targeting calcium as single ion might be too simplistic. Goldberg and colleagues demonstrated in experiments that if only calcium is removed, the neuronal damage is enhanced. Only if calcium, sodium, and potassium are removed from the culture medium, protection is achieved. The group concluded that there are two phases to neuronal ischaemic damage pathways: the first one caused by acute swelling, the second one depending on the rise in intracellular calcium concentrations [23]. Thus, there are calcium-independent effects. Regarding these publications, it seems reasonable to conclude that nimodipine does play a role in the calcium metabolism of an ischaemia-exposed neuron; however, the L-type calcium channel alone is likely not central enough in the complex system of calcium-mediated cell death to act as a promising drug target. Our data show no evidence of a protective effect against OGD in our hippocampal slice culture model. This points towards a minor role for L-type calcium channels within the pathophysiological cascade of early ischaemic cell damage.

### 3.2. Amiloride and Combined L-Type Calcium Channel and ASICs Inhibition

ASICs have been identified to be accountable for increased levels of intracellular calcium in ischaemic conditions. Under acidic conditions, as it occurs during ischaemia, ASIC1a channels become permeable for calcium and provide a mechanism for calcium influx that is not dependent on depolarisation or excitotoxicity [29]. In our study, the ASIC channel inhibitor amiloride did not show any significant neuroprotective effect. This contrasts with other studies showing a protective effect of ASIC inhibition on brain injury [27]. Positive effects of ASIC inhibition were most prominent in a combined acidosis and OGD model. That may explain the insufficient protection of amiloride in our pure OGD model in which the OGD induced acidosis may be less pronounced. Besides, a combination of L-type calcium channel and ASIC inhibition by nimodipine plus amiloride did also not show a protective effect in our model. These findings point against a significant damaging role of calcium entry in parenchymal cells via VGCCs and ASICS during OGD in models not depending on an intact blood supply. In our study, the amount of damage was investigated 72 h after the insult. It can therefore not be ruled out that an early, only transient, and thus not sustained effect of calcium entry blockade via VGCCs and ASICs may have occurred.

Interestingly, in slices treated with regimes using the possibly harmful concentration of 2.0 Vol.% DMSO, we observed a significant further increase in cell damage only for slices where nimodipine and amiloride were applied in combination. DMSO at this high concentration already caused damage in our control slices without OGD. It is also known from the literature that DMSO higher than 5% may have harmful effects on biological tissue and cells [44,45]. Safe concentrations are described up to 3% in hippocampal neurons [44]. In combination with OGD, 2 Vol.% DMSO did not induce an increase in damage beyond the OGD induced damage. In addition, slices receiving DMSO at this high concentration with additional 100µM amiloride did not significantly differ from the OGD control group. We did not investigate nimodipine alone in a concentration of 2.0 Vol.% DMSO. Thus, we cannot draw conclusions whether this effect is caused by the combination of amiloride and nimodipine in 2.0 Vol.% DMSO, or whether nimodipine combined with high concentration of DMSO alone would have the same effect. The high DMSO concentration of 2 Vol.% may have affected the system in addition to OGD, inducing a setting of higher damage-susceptibility, where calcium channel inhibition (equal whether via nimodipine alone or in combination with amiloride) is even harmful instead of neuroprotective. If we assume a combined mechanism, impairment of regulatory systems responsible for ASIC function by the simultaneous L-type calcium channel inhibition with nimodipine might be the answer. This could either be caused by a reduction in intracellular calcium or by the missing reduction in extracellular calcium. Paukert et al., for example, showed that extracellular calcium influences ASIC function and can be competitively inhibited by calcium ions. Furthermore, they suggest that higher extracellular calcium shifts ASIC sensitivity to more acidic pH levels [33]. Admittedly, we would rather expect less ASIC activity and thus less damage if we follow this argumentation. A further explanation may be based on amiloride’s unspecific effects. Amiloride does also interfere with other channels, e.g., Na^+^/H^+^- or Na^+^/Ca^2+^-exchangers and even T-type calcium channels [46]. As aforementioned, calcium extrusion mechanisms appear to be crucial for neuronal integrity. It could be imagined that unspecific blockage mechanisms in combination with L-type inhibition prevent calcium homeostasis mechanisms in the damaging environment of possible toxic effects of DMSO and OGD. Lastly, the concentration of 2.0 Vol.% DMSO is a comparably high choice for the solvent. We recommend avoiding this concentration if feasible.

### 3.3. Nimodipine—Most Important Effect on the Vasculature

Despite years of research, nimodipine is the only drug available that shows an improved functional outcome and positive effects on mortality after SAH. However, the exact underlying mechanism is still unclear [40]. As pointed out in this publication, a direct neuroprotective effect appears to be unlikely. After the CONSCIOUS-1 study [47], the scientific community changed their perspective on vasospasms of large vessels and DCI. The latter is now considered a complex condition with possible pathways in micro-thrombi [48,49], micro-vasculature spasms (mostly described in experiments and possibly a cause for the secondary development of micro-thrombi) [50,51,52], neuroinflammation [53,54], and cortical spreading depolarization [55]. Indeed, there is evidence from the literature that nimodipine interacts with several of the abovementioned mechanisms rather than directly acting on neurons, finally resulting in neuroprotection and better functional outcome. That explains a lack of neuroprotection in our model using organotypic slice cultures.

### 3.4. DMSO as Solvent

DMSO is a commonly used solvent in slice experiments similar to the one presented here. In healthy slices, DMSO applied for 72 h displayed significant damage to the slices at the highest concentration of 2 Vol.% tested. The lower concentrations only induced a concentration-dependent tendency towards slightly enhanced but not relevant damage. In contrast to this effect on slices without OGD, while applying DMSO following OGD, 1.0 Vol.% DMSO alone appears to induce a slight, although not statistically significant, reduction in cell damage. The highest concentration of 2 Vol.% DMSO did not show a protective effect but also did not significantly add to the OGD induced damage. This suggests identical damage pathways for DMSO at this high concentration and OGD. DMSO seems to have a U-shaped concentration-dependent effect on slices after OGD. Neuroprotective effects through DMSO have been described in different experimental designs and models [44,56,57], and thus, DMSO was even suggested as a treatment option for ischaemic brain conditions [58]. Lu and Mattson, e.g., report that DMSO inhibits the glutamate-induced (excitotoxic) calcium influx in hippocampal rat neurons at DMSO levels from 0.5 to 2.0% [44]. Suppression of excitotoxicity may thus also be a possible explanation for our observation of slightly protective effects at 1.0 Vol.%, since neuronal connections stay intact in organotypic hippocampal slices [59]. Another suggested explanation is DMSO’s property as a scavenger for free oxygen radicals [58]. However, in our control slices, we observed a dose-depending increase in cell damage with significantly more damage for DMSO at 2.0 Vol.% compared to slices without DMSO or with 0.1 Vol.%. Contrary to the inhibiting effect on excitotoxicity, Galvao et al. have reported apoptosis-inducing effects in retinal cells using DMSO concentrations as low as 1% [60]. Zhang et al. confirmed these observations in experiments with neurons and astrocytes, reporting neuronal alterations at 0.5% DMSO [61]. In addition, even the smallest DMSO concentrations are described to have effects on cell metabolism, and the solvent possibly accumulates in brain slices [62]. From these diverse literature findings, a clear concentration-dependent effect—regardless of protective or harmful—cannot be identified. We consider it an important technical note to this paper that for brain slice experiments addressing mechanisms of neuroprotection, DMSO should be avoided if feasible. If DMSO is utilised, it should only be done so with the utmost caution and proper utilisation of vehicle controls.

### 3.5. Quality of Slices—Need for Defined Exclusion Criteria

In our group, we recognised that in many publications using organotypic hippocampus slices the inclusion criteria for slices are unclear or not exactly defined. Organotypic slice cultures are highly sensitive to external influences and can therefore be inhomogeneous with regard to cell damage [63]. We claim that a standardised inclusion pipeline would benefit further research conducted using this method. With the here-suggested pipeline (see Materials and Methods), we are confident that we enhanced the quality of data used for statistical analysis in our experiments.

### 3.6. Limitations

There are several limitations that may be considered given our conclusions:

Firstly, the described interactions of DMSO in the concentrations used may have interfered with the neuroprotective effects of both tested substances. However, this may only account for the highest concentration of 2.0 Vol.% of DMSO and not for the lower concentrations, not showing significant effects on the outcome in control as well as OGD. Secondly, we did not use a specific imaging technique to depict calcium concentrations. By applying well-established concentrations of nimodipine and amiloride, we are confident that an effective channel blockade was achieved, preventing calcium influx via these ion channels. Thirdly, we planned smaller groups for vehicle controls and control groups from the beginning to minimise the number of required animals. Thus, some control groups have limited sample size compared to treatment groups, and further power would be eligible regarding our observation of a possible neuroprotective effect of DMSO at a concentration of 1.0 Vol.%. Lastly, we did not investigate the effect of nimodipine in 2.0 Vol.% DMSO. Hence, it is not possible to draw specific conclusions from our observation of enhanced damage of combined treatment with nimodipine and amiloride at 2.0 Vol.% DMSO.

## 4. Materials and Methods

### 4.1. Mediums

Preparation medium (Gey’s balanced salt solution (Sigma-Aldrich, Munich, Germany), 5 mg/mL D-(+)Glucose (Roth, Karlsruhe, Germany)) [63] was used for initial slice manufacturing. The growth medium used for slice culturing consisted of 50% Eagle minimal essential medium with Earle‘s salts (Sigma-Aldrich), 25% Hank’s balanced salt solution (Sigma-Aldrich), 25% heat inactive horse serum (Sigma-Aldrich) with additional 5 mg/mL D-(+)Glucose (Roth, Karlsruhe, Germany), 1 Vol.% antibiotic/antimycotic solution [penicillin G GIBCO™, 10,000 units/mL, streptomycin sulphate 10 mg/mL, amphotericin B 25 µg/mL] (Thermo Fisher Scientific, Waltham, MA, USA), 5 µL/mL medium L-Glutamine solution (Sigma-Aldrich) and 10 µL/mL HEPES buffer solution (Sigma-Aldrich) [63]. For experiments, the experimental medium (75% Eagle minimal essential medium with Earle‘s salts (Sigma-Aldrich), 25% Hank’s balanced salt solution (Sigma-Aldrich) with additional, 5 mg/mL D-(+)Glucose (Roth, Karlsruhe, Germany), 1 Vol.% antibiotic/antimycotic solution [penicillin G GIBCO™, 10,000 units/mL, streptomycin sulphate 10 mg/mL, amphotericin B 25 µg/mL] (Thermo Fisher Scientific, Waltham, MA, USA), 5µL/mL L-Glutamine solution (Sigma-Aldrich) and 10µL/mL HEPES buffer solution (Sigma-Aldrich)) or OGD medium (75% Eagle minimal essential medium with Earle‘s salts (Sigma-Aldrich), 25% Hank’s balanced salt solution (Sigma-Aldrich) with additional 1 Vol.% antibiotic/antimycotic solution [penicillin G GIBCO™, 10,000 units/mL, streptomycin sulphate 10 mg/mL, amphotericin B 25 µg/mL] (Thermo Fisher Scientific, Waltham, MA, USA), 5µL/mL L-Glutamine solution (Sigma-Aldrich) and 10 µL/mL HEPES buffer solution (Sigma-Aldrich)) were used, respectively. In essence, OGD medium is simply experimental medium without D-(+)Glucose.

### 4.2. Slice Preparation and Cultivation

The experiments in this article were strictly conducted according to institutional and governmental guidelines (TierSchG) with institutional permission by the animal protection representative of the Institute of Animal Research at the RWTH Aachen University Hospital and the local institutional committee (LANUV North Rhine-Westphalia, TV-11141A4). After decapitation of 4–7-day-old mice pups (C57BL/6N from Charles Rivers Laboratories, Sulzfeld, Germany and from Janvier Labs, La Rochelle, France, *n* = 138), their brains were extracted and instantaneously immersed into ice-cold preparation medium. The hippocampus slices were prepared using an already established method [58,63]. In brief, the brains were sagittally divided in half, and the frontal pole, as well as the cerebellum, was resected. Using a McIlwain Tissue Chopper (The Mickle Laboratory Engineering Co. ltd. [Now: Cavey Laboratory Engineering Co. ltd], Gomshall, UK), the brains were sliced into 400 µM thick slices from which the hippocampus was then carefully dissected. The hippocampus slices were then transferred onto MilliCell tissue culture inserts (MilliCell-CM, Millipore Corporation, Billerica, MA, USA) placed in 1 mL growth medium. Slices were cultivated at 37 °C and 5% CO2 for 14 days with the growth medium exchanged one day after the preparation and every following third day. On average, 9.1 slices per pup were prepared. Slices of one animal were allocated in two wells, which were then randomly allocated to the experimental groups to avoid allocation of slices of the same animal to only one experimental group. Overview in Figure 6a.

### 4.3. Imaging

To obtain baseline images, immediately before the OGD experiment growth medium was exchanged for the experimental medium with additional 3 µL/mL propidium iodide (PI) and then incubated again for a minimum of 30 min at 37 °C and 5% CO_2_. Propidium iodide stains DNA of cells with impaired cell membrane and was used to assess the number of cells damaged [64] using a fluorescence microscope (Zeiss Axioplan, Carl Zeiss MicroImaging GmbH, Jena, Germany) (exposure time was calculated for every imaging session and typically ranged between 15.500 and 16.500 milliseconds) and MetaVue software (MetaVue, Molecular Devices, Sunnyvale, CA, USA). Imaging was performed at baseline for all slices and 72 h after experiments. Imaging is depicted in Figure 6a,c.

### 4.4. Oxygen–Glucose Deprivation (OGD)

For OGD (Figure 6b), first OGD medium was aerated with 95% N_2_, 5% CO_2_ for 30 min using a Spectron flowmeter FLM-32 (Spectron Gas Control Systems GmbH, Frankfurt, Germany) at a rate of 15% at 0.2 bar to desaturate the OGD medium from oxygen and warmed up afterwards. The medium was quickly exchanged with OGD medium for the OGD groups and normal experimental medium for control groups, respectively, before immediately being transferred into air-tight experimental chambers (750 mL volume). Chambers containing OGD slices were then flushed with 95% N_2_, 5% CO_2_ at a rate of 100% at 0.5 bar for 6 min (resulting in a flow of 2.73 L/min) to guarantee a sufficiently hypoxic environment [35]. The chambers were then sealed, and OGD was sustained for 60 min to receive a reasonable amount of cell damage [23]. After OGD, all slices were changed back to experimental medium with additional 3 µL/mL propidium iodide (PI) and randomly allocated to neuroprotective protocols. Thereafter, the slices were again incubated at 37 °C and 5% CO_2_ for 72 h.

### 4.5. Neuroprotective Protocols

Neuroprotective drugs were dissolved using dimethyl-sulfoxide (DMSO) (Sigma-Aldrich, St. Louis, MO, USA) [65]. Vehicle controls were performed for OGD and control groups for 0.1 Vol.%, 1.0 Vol.% and 2.0 Vol.% DMSO. For OGD groups, nimodipine was applied at 10 or 20 µM dissolved in either 0.1 Vol.% or 1.0 Vol.% DMSO and in combination with amiloride at 100 µM [27] again at 10 or 20 µM with 1.1 Vol.% or 2.0 Vol.% DMSO, respectively. Finally, we applied amiloride at 100 µM alone with either 1.0 Vol.% or 2.0 Vol.% DMSO. All agents remained within the medium for 72 h after OGD or time control, ending the experiments with the final imaging. An overview of all protective protocols can be seen in Figure 6b and in Appendix A.

### 4.6. Cell Death Assessment

PI resulted in red staining of damaged cells. To assess the amount of cell death within each slice, we used python to split channels for each picture into red, green, and blue and then created a corresponding picture in grey values for the red channel only. Again, using python, histograms were created for each picture depicting the corresponding grey-scale values from 0 to 255. A threshold for pixels below a grey-scale value of 100 was used to filter background fluorescence [35,63,66]. Finally, all pixel values were summed to resulting in one value per pixel representing the total damage for each slice.

### 4.7. Pre-Statistical Image Processing

To further process the slices for total damage analysis, the following exclusion criteria were defined. Every slice was excluded when [1.] more than one slice was shown on the corresponding images at 72 h (*n* = 6 excluded), when [2.] the dentate gyrus was not reliably identifiable (*n* = 57 excluded), when [3.] the CA1-region was not reliably identifiable (*n* = 28 excluded), when [4.] we identified unexplained dark spots in the picture (*n* = 31 excluded), when [5.] PI clots were observed on the image (*n* = 18 excluded), when [6.] slices were not plane but exhibited “wrinkling” that likely occurred in the cultivation process (*n* = 78 excluded), or when [7.] slices presented an inhomogeneous margin, likely due to inadequate preparation technique (*n* = 67 excluded). In total, 227 slices were excluded, and 1032 slices were used for further analysis. It should be noted that several slices showed more than one of the abovementioned features. We provide examples and possible explanations for our observations in the Appendix B.

In addition, to only include slices showing no preparation and cultivation induced damage already before OGD, we specified a maximal pre-damage threshold at the mean plus one standard deviation of all 1032 slices so far identified as useful. Seventy-four slices at 0 h presented a level of pre-experimental damage that was above that threshold. Finally, we defined a further threshold for the slices at 72 h after OGD to obviate extreme outliers in each of the experimental groups. For each of our 20 experimental groups, we established an individual threshold of the mean plus two times the standard deviation as a maximal damage plausibly caused by our OGD method (this excluded 33 slices at 72 h). By applying these criteria, comparability between slices was enhanced, as pre-damaged slices were restrained from entering the experiment and unrealistic outliers were excluded. The experimenter applied all exclusion criteria blinded to the slice allocation to the experimental groups. In total, 334 slices were eliminated, and 925 slices were included in our final analysis.

### 4.8. Statistical Analysis

Statistical Analysis was performed using GraphPad Prism version 8.2.0 for Windows (GraphPad Software, San Diego, CA, USA). Our data were normalised with the mean of the untreated OGD group as reference. Using a Kolmogorov–Smirnov test, we concluded that the assumption of a normal distribution was not met by our data. A two-tailed Mann–Whitney test between our control and our OGD group was used to verify our OGD model caused adequate damage. We used the Kruskal–Wallis test comparing the mean of rank of each group with every other group between the 72 h data of all OGD groups and for our control groups, respectively. Dunn’s test was used to correct for multiple comparisons. For all statistical analysis, a *p*-value <0.05 was considered significant. To calculate Cohen’s D, a short Python script was used.

## 5. Conclusions

The cellular mechanisms of neuronal ischaemic damage are incredibly complex and effective neuroprotection is—as desired as it might be for all professionals working in that field—still a long way down the road. The idea of only targeting one of the mechanisms, while ischemia activates a symphony of potentially harmful pathways, is probably too simplistic. Nevertheless, the role of calcium is crucial, and the interplay of different calcium channels and their respective effect on cell injury needs further research efforts. Nimodipine remains a hot topic in SAH research mainly due to its vascular effect. Based on encouraging recent findings of a possible calcium-independent effect of nimodipine on microglia [67], we suggest to additionally add a perspective of possible neuro-regeneration to the list of experimental questions, as well as further observing effects on the microvasculature.

We again want to stress that in vitro experiments using DMSO as a solvent should be evaluated critically, since the popular drug vehicle interacts with neuronal damage mechanisms.

## Figures and Tables

**Figure 1 ijms-21-08921-f001:**
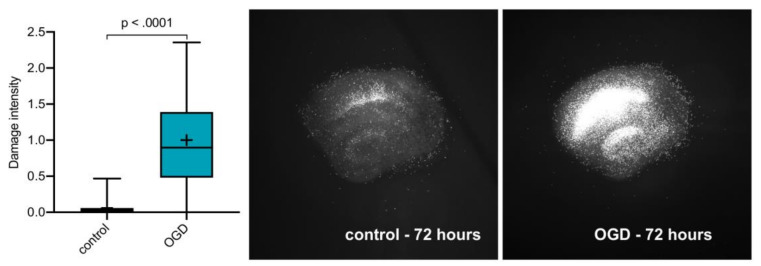
Damage intensity of the control group (*n* = 101) vs. oxygen–glucose deprivation (OGD) group (*n* = 96). Mann–Whitney U test was used to determine whether our OGD model caused adequate damage. Slice images shown are example slices for control and OGD groups, both at 72 h with typical means for the grey-scale value. Slice images are red-channel-filtered and contrast-enhanced.

**Figure 2 ijms-21-08921-f002:**
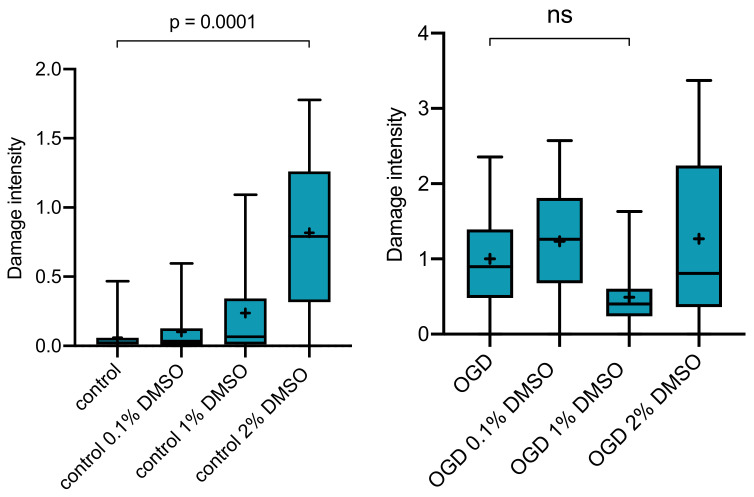
(**Left**): Slices receiving 2.0 Vol.% dimethyl-sulfoxide (DMSO) alone (*n* = 20) without undergoing OGD showed significantly (*p* = 0.0001) more cell damage. This effect can be observed for 0.1 (*n* = 25) and 1.0 Vol.% (*n* = 12) as well but at a much lower level without statistical significance. (**Right**): No significant effects of DMSO in slices undergoing OGD were observed for 0.1 (*n* = 41), 1.0 (*n* = 19) or 2.0 (*n* = 45) Vol.% DMSO compared to the OGD group without DMSO (*n* = 96). The figure depicts a tendency towards less cell damage at 1.0 Vol.% DMSO.

**Figure 3 ijms-21-08921-f003:**
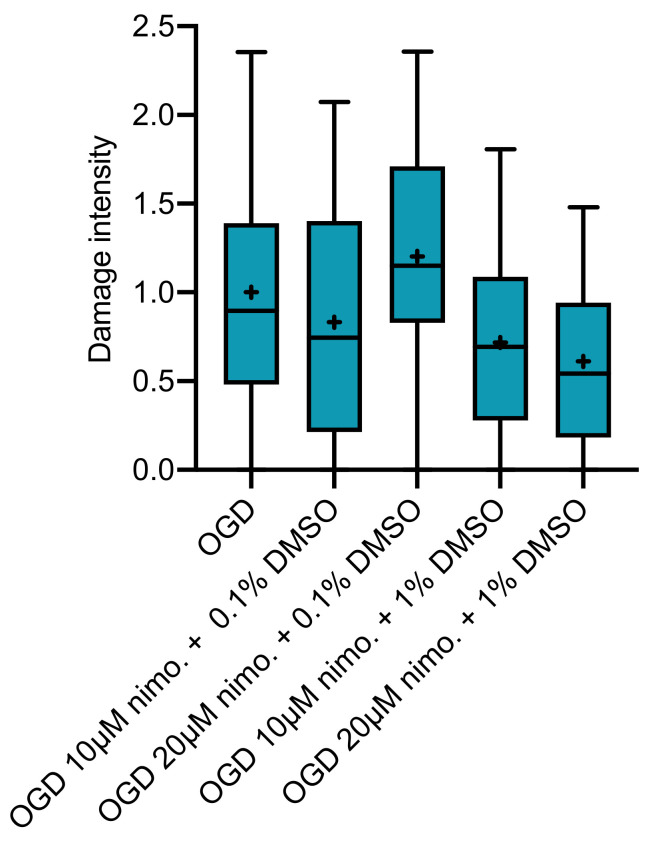
Slices undergoing OGD received nimodipine at either 10 or 20 µM dissolved in 0.1 or 1.0 Vol.% DMSO. No significant differences in cell damage were observed at 72 h after OGD (OGD *n* = 96; dissolved in 0.1 Vol.% DMSO: 10µM nimodipine *n* = 77, 20 µM nimodipine *n* = 64; dissolved in 1.0 Vol.% DMSO: 10µM nimodipine *n* = 30, 20 µM nimodipine *n* = 25).

**Figure 4 ijms-21-08921-f004:**
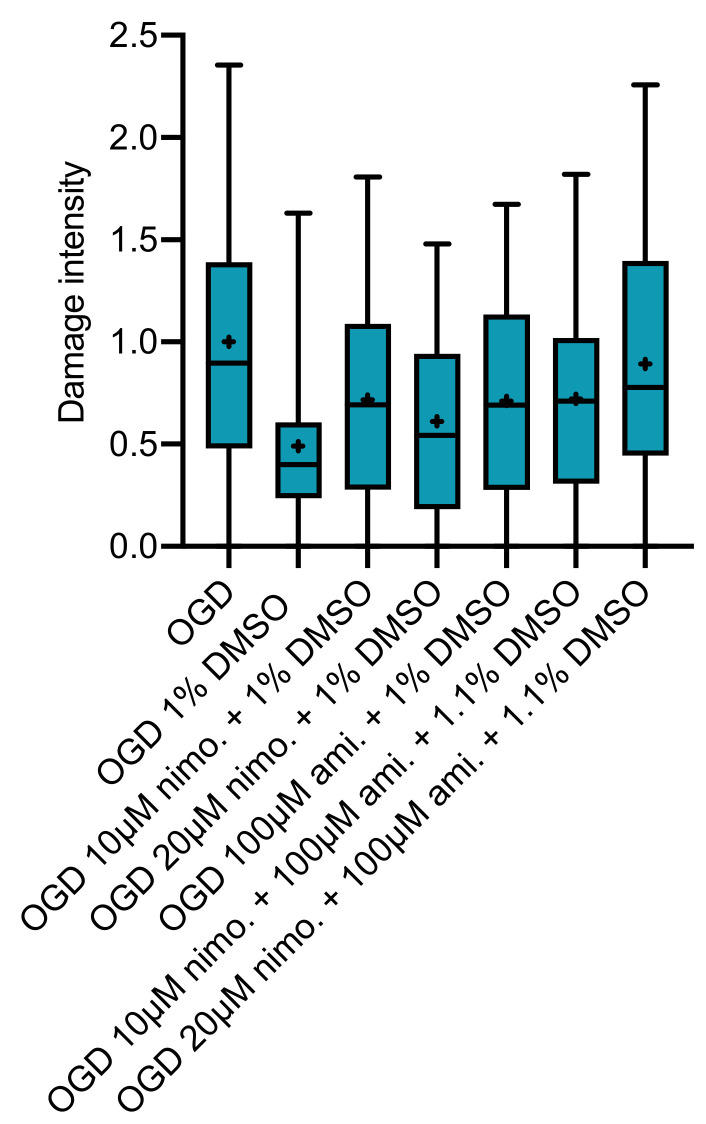
Slices receiving 10 (*n* = 30) or 20 µM (*n* = 25) nimodipine in 1.0 Vol.% DMSO compared to slices receiving 100 µM amiloride alone in 1 Vol.% DMSO (*n* = 50) or in combination with 10 (*n* = 56) or 20 µM (*n* = 59) nimodipine in 1.1 Vol.% DMSO. No significant difference in cell damage intensity was observed at 72 h after OGD.

**Figure 5 ijms-21-08921-f005:**
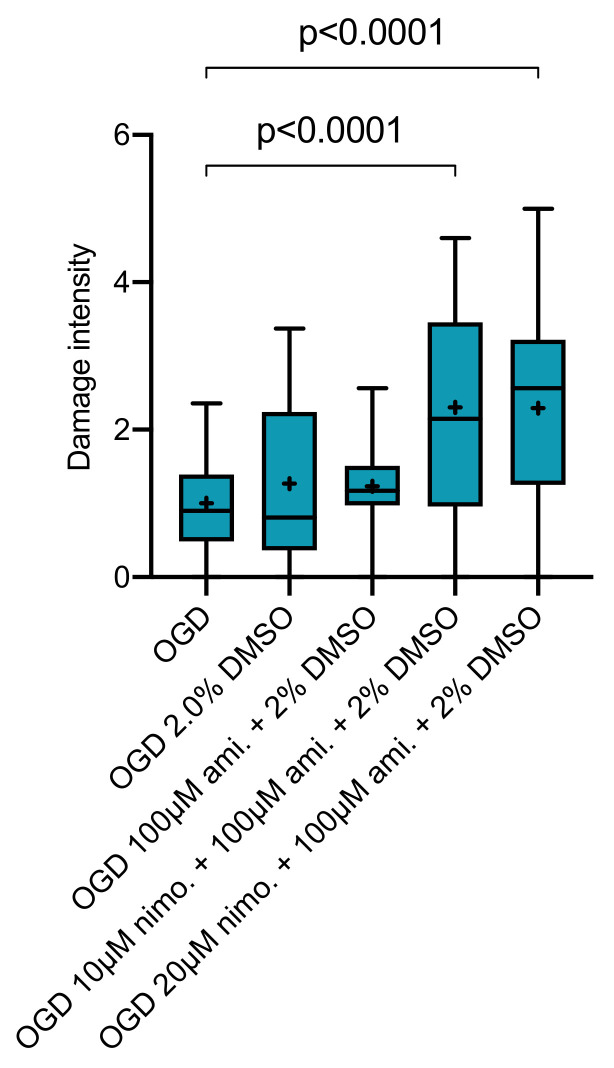
Slices receiving a combination of either 10 (*n* = 47) or 20 µM (*n* = 46) nimodipine with 100 µM amiloride in 2.0 Vol.% DMSO showed significant impairment of cell viability at 72 h (Kruskal–Wallis test with post hoc Dunn’s test, *p* < 0.0001) compared to OGD slices (*n* = 96). This effect was not observed for slices receiving 100 µM amiloride in 2.0 Vol.% DMSO (*n* = 43) or 2.0 Vol.% DMSO alone (*n* = 45).

**Figure 6 ijms-21-08921-f006:**
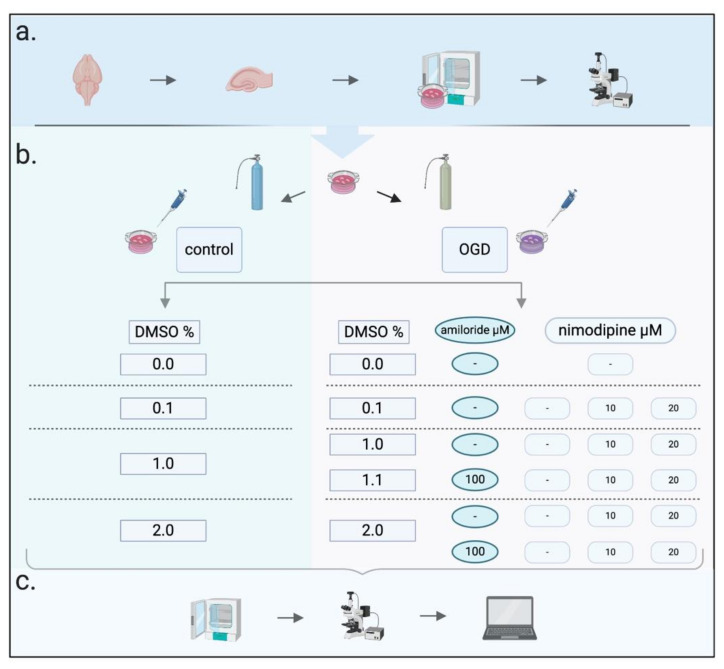
Experimental design. (**a**) Slice preparation, culturing for 14 days and baseline propidium iodide (PI) imaging. (**b**) OGD experiments with overview of experimental groups. (**c**) Incubation for 72 h after OGD, PI imaging for cell death assessment at 72 h, analysis of experimental groups. Created with BioRender.com.

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
