# Peer review of "Failed Neuroprotection of Combined Inhibition of L-Type and ASIC1a Calcium Channels with Nimodipine and Amiloride"

_ijms, 2020, doi:10.3390/ijms21238921_

Round 1
Reviewer 1 Report
My concerns were adequately answered.
Please, there is a typo in line 342. "Depicted"
Author Response
My concerns were adequately answered.
The authors appreciate the reviewer's comment.
Please, there is a typo in line 342. "Depicted"
We thank the reviewer for his comment. This misspelling has been corrected in the revised manuscript.
Reviewer 2 Report
The manuscript has been improved, and most of my concerns have been addressed.
Author Response
The manuscript has been improved, and most of my concerns have been addressed.
We again appreciate the reviewer's comments and remarks that helped us to improve the manuscript.
This manuscript is a resubmission of an earlier submission. The following is a list of the peer review reports and author responses from that submission.
Round 1
Reviewer 1 Report
The manuscript titled as ‘Failed Neuroprotection of Combined Inhibition of L-Type and ASIC1a Calcium-channels with Nimodipine and Amiloride’ by Ort et al. investigated two clinically used drugs – nimodipine and amiloride through an in vitro experimental model of ischaemia, and found neither nimodipine (at 10 or 20 µM) alone or in combination with amiloride (at 100µM) showed any amelioration. This work could be of interests to researchers on neuroscience and molecular medicine etc.
However, as the authors have realized that nimodipine has been reported to improve functional outcome in ischaemic and haemorrhagic stroke in in-vivo animal experiments as well as in clinical trials [13,18,20,37,38], whereas, in this manuscript, there is such drug effects as the authors investigated their work through an in-vitro model of oxygen-glucose deprivation (OGD) in organotypic hippocampus slices of mice. Three is still a missing control (positive control) for this in vitro model, which should be investigated in parallel with DMSO, nimodipine and amiloride, to make a solid conclusion. Are there any compounds, as positive controls, to prevent the intracellular rise of calcium concentration in this in-vitro model of OGD?
Reviewer 2 Report
The authors present a complex set of in vitro pharmacological assays with negative results (no treatment effects). They demonstrated no neuroprotective effects of different class of calcium+2 channel blockers in organotypic preparation of hyppocampal slices submitted to ischemic insult. To measure cell death, the authors used DAPI, which stains nuclei of dying cells.
My major concern is the complex mix of solvent and calcium channel blockers effects. Both drugs and solvent trigger cell death in a non-linear fashion (U-shape in the case of DMSO, non-dose dependent in the case of nimodipine). In addition, the are also reports about the interference of DMSO in calcium+2-mediated glutamate excitotoxicity.
The main weakness of the study is the figure 5. If one compares it to figure 4, there is a reduction of treatments from 7 to 5, which means that the 10 and 20 microM of nimodipine in 2% of DMSO groups are missing. Without these experimental/treatment groups, it is not possible to discern the additive effects of nimodipine and amiloride. This flaw invalidates the main conclusions of the study.
The hypothesis that cell death in ischemia is not all everything about intracellular overload of Ca+2 constitutes a very interesting research pursuit. But it is hard to understand the whole set of results. Another flaw is the measurement of cell death itself. The authors rely on just one method. Why not to use an counterstaining for living cells? Might high DMSO levels interfer with DAPI staining? A 2% of DMSO is too much.
Also I missed some depiction of the ANOVAs (if any).